# Knowledge, practice, and impact of COVID-19 on mental health among patients with chronic health conditions at selected hospitals of Sidama regional state, Ethiopia

**Yilkal Simachew**[1]*, **Amanuel Ejeso**[2], **Sisay Dejene**[1], **Mohammed Ayalew**[3]

**1** School of Public Health, College of Medicine and Health Science, Hawassa University, Hawassa, Ethiopia, **2** Department of Environmental Health, College of Medicine and Health Sciences, Hawassa University, Hawassa, Ethiopia, **3** College of Medicine and Health Science, School of Nursing, Hawassa University, Hawassa, Ethiopia

* joemakalister123@gmail.com

**Data Availability Statement:** All relevant data are within the paper and its Supporting Information files.

## Abstract

### Background

COVID-19 causes worse outcomes and a higher mortality rate in adults with chronic medical conditions. In addition, the pandemic is influencing mental health and causing psychological distress in people with chronic medical illnesses.

### Objective

To assess the knowledge, practice, and impact of COVID-19 on mental health among chronic disease patients at selected hospitals in Sidama regional state.

### Method

A facility-based cross-sectional study was conducted. A total of 422 study subjects were enrolled in the study using a two-stage sampling technique. Data were coded and entered using Epi Data version 3.1 and exported to SPSS-20 for analysis. Descriptive analysis was used to present the data using tables and figures. Bivariate and multivariate logistic analyses were used to identify factors associated with the initiation of preventive behavior of COVID-19. Variables with a P-value of less than 0.25 in bivariate analysis were considered as candidate variables for multivariable analysis. The statistical significance was declared at a P-value less than 0.05.

### Result

More than half 237 (56.2%, 95% CI: 50.7–60.9) of the study participants had good knowledge of COVID-19. The practice of preventive measures toward COVID-19 was found to be low (42.4%, 95% CI: 37.9–47.2). Being widowed (AOR = 0.31, 95% CI (0.10, 0.92)), secondary and above educational status (AOR = 2.21, 95% CI (1.01, 4.84)), urban residence (AOR = 2.33, 95% CI (1.30, 4.19)) and good knowledge (AOR = 4.87, 95% CI (2.96, 8.00))

**Funding:** The study was funded by Hawassa University. The funder had no role in study design, data collection and analysis, decision to publication, or preparation of the manuscript.

**Competing interests:** The authors have declared that no competing interests exist.

were significantly associated with good practice. In addition, more than one-third of the study participants 37% (95% CI 32.7, 41.5) were experiencing anxiety. While more than a quarter of respondents 26.8% (95% CI 22.5, 31.5) had depression.

## Conclusion and recommendation

Despite more than half of the participants had good knowledge, the prevention practice was low. Hence, multiple information dissemination strategies should be implemented continuously among chronic disease patients. In addition, the magnitude of concurrent depression and anxiety in the current study was high.

## Introduction

Since December 2019, the global population is in health crisis because of the COVID-19 pandemic caused by Severe Acute Respiratory Syndrome Corona virus-2 (SARS-COVID-2) which has been identified in Chinese patients with severe pneumonia and flu-like symptoms [1]. COVID-19 is a contagious disease that spreads rapidly via droplets from the sneezing and coughing of an infected person. The disease is highly contagious and has a 14-day incubation period. Its main clinical symptoms are fever, dry cough, sneezing, sore throat, headache, bodily pain, fatigue, chills, and shortness of breath [2].

According to the World Health Organization (WHO) report issued in February 2021, more than 108 million confirmed COVID-19 cases and more than 2.3 million deaths were reported globally [3]. From this figure, more than 2.7 million confirmed cases and over 68 thousand deaths have been reported in the African region. Ethiopia is the third country where deaths due to COVID-19 are high in the regions of Africa, next to South Africa and Algeria [4]. Ethiopia is currently suffering from the critical phase of COVID-19, so there is concern that the infection will spread rapidly.

Not all people are equally affected by the virus; people with chronic medical illnesses were more likely to become infected [5]. In addition, the COVID-19 infection has a worse outcome and a high mortality rate among this group of people [6]. Individuals living with chronic diseases are already viewed as being among the most greatly impacted by stressors related to the COVID-19 pandemic. These underscore the importance of preventive measures (such as social distancing, respiratory hygiene, and wearing a face mask in public) in protecting people with chronic medical conditions. To improve preventive behaviors among these vulnerable groups, it is important to assess their knowledge and action related to COVID-19 and identify the determinants that affect the initiation of preventive behaviors [7].

As the coronavirus pandemic spreads quickly around the globe, it creates a great deal of fear, worry, and concern in the public at large, and among certain groups in particular, such as care providers, older adults, and people with underlying health conditions [8, 9]. In addition, from the past experience, such a pandemic is known to cause beyond a physical illness, it also has an impact on people's mental health and psychological well-being, particularly those with underlying medical conditions. Due to the severity of the disease among these vulnerable groups, they develop psychiatric disorders such as depression and anxiety during these health crises, which can impede infection control [8]. That is why this study wants to measure the mental health impact of the COVID-19 pandemic among these vulnerable groups.

Ethiopia has made a strong commitment to prevent and slow down the COVID-19 pandemic before it causes a major public health crisis [10]. WHO recommends that apart from

case identification, contact tracing, quarantine, and large-scale screening in vulnerable groups; controlling the spread of disease among high-risk groups must be one of the strategies of COVID-19 prevention practice [11]. Despite this, the majority of studies in Ethiopia focused on health professionals and the general population, rather than the vulnerable groups of chronic disease patients. The result of this study may help to direct the efforts and plans of public health authorities and media of the country for better and timely control of COVID-19. Thus, this study aimed to determine the knowledge and practice toward COVID-19, the prevalence of depression and anxiety during the COVID-19 pandemic, and the associated factors of good knowledge and practice among patients with chronic disease at 3 hospitals in the Sidama region.

## Methods and materials

### Study area

The study was conducted in selected public hospitals in Sidama regional state. Sidama region is one of the regional states of Ethiopia with a population of more than 4 million, where more than 90% of the population lives in the rural part of the region. It has a total of 30 districts and 6 town administrations. It is located 272 km to the southeast of Addis Ababa and the region has a total area of 6,981.8 square kilometers. According to a 2018 estimate, the total population of the region was 4,294,730 of which 2,104,418 are females and 2,190,312 are males [12]. Currently, there are 16 public hospitals, 126 health centers, 531 health posts, 24 medium and 83 primary privates, and 7 NGO clinics [13].

### Study design and population

A facility-based cross-sectional study was conducted from June 20 to July 30/2020. The source population of this study was all patients with chronic diseases (hypertension, cardiovascular disease, diabetes, chronic respiratory disease, and chronic kidney disease) who attended the chronic disease follow-up clinic at three hospitals in the Sidama region. While the study population was all patients with chronic disease who attended the chronic disease follow-up clinics at three hospitals during the study period.

### Inclusion and exclusion criteria

All chronic disease patients who were on follow-up at the selected hospitals during the study period were included. The study excluded chronic disease patients who were severely ill, had cognitive impairment, were younger than 18 years old, and health professionals.

### Sample size determination

The minimum sample size required for the objectives of the study was calculated by using the single population proportion formula, using the following assumptions

$$n = \frac{(Z\alpha/2)^2 p(1-p)}{d^2}$$

n = minimum sample size

Z = Critical value for normal distribution at 95% confidence level which is 1.96 (Z value at α = 0.05, two-tailed)

P = Due to the lack of other studies in a similar setting, we used the anticipated population proportion of 50%.

d = margin of error to be tolerated (5%)

Sample size = 384
After considering a 10% non-response rate, the final sample size was 422

## Sampling procedure

The Sidama regional state has a total of 16 hospitals (2 general hospitals, 13 primary hospitals, and 1 comprehensive specialized hospital). Based on patient flows, 3 hospitals were selected purposively, and a sampling frame was prepared for each hospital based on the number of chronic disease patients in the last month who attend the chronic disease follow-up clinics. Based on one month of data, there were around 1376, 1100, and 900 chronic disease patients who attend the chronic disease follow-up clinic at Hawassa university comprehensive special-ized hospital, Yirgalem general hospital, and Leku primary hospital respectively. A propor-tional sample to the number of patients at each hospital was allocated, and then the systematic sampling technique (K = 8) was used to select each study participant from their respective group.

## Study variables

The dependent variables were knowledge and practice toward the COVID-19 pandemic. The independent variables were socio-demographic variables such as sex, age, place of residence, educational status, occupation, marital status, monthly income etc. . .

## Data collection procedure and tool

Data was collected using a structured interviewer-administered questionnaire. The question-naire was adapted from the WHO website and other relevant COVID-19 literature [14, 15]. The questionnaire consisted of four parts (see S1 File). The first part included the socio-demo-graphic characteristics of the study participants. The second part included 16 questions designed to assess the knowledge of COVID-19. The knowledge questions were answered with options Yes/No/I don't know. In the third part, 15 questions with Yes/No answer options were used to assess practice.

Depression and anxiety were measured by the Hospital Anxiety and Depression Scale (HADS). This tool contains 14 items (seven for each) that are scored in Likert form from 0 to 3; giving a total of 21 points. The scale has been validated in different populations in Ethiopia [16]. Data were collected using 5 diploma nurses and supervised by three supervisors with bachelor's degrees in public health.

## Data analysis procedure

The collected data were entered into Epi data version 3.1 and exported to the statistical pack-age for social science (SPSS) version 20 for analysis. First descriptive analysis was carried out for each of the variables (including for the outcome variable knowledge and practice). A cor-rect answer (Yes) for the knowledge question was assigned 1 point, and No/I don't know answer was assigned 0 points. The total knowledge score ranged from 0 to 16, Participants' overall knowledge was categorized using Bloom's cut-off point as Good if the score was between 80 and 100% (12.7–16 points), and Poor if the score was less than 79% (<12. 6 points) [15]. The correct answer for the practice question was assigned 1 point and an incorrect answer was assigned 0 point. The total practice score range from 0 to 15. The overall practice score was categorized using Bloom's cut-off point, as Good if the score was between 80 and 100% (12–15 points) and Poor if the score was less than 79%(< 11.9 points) [15].

HADS tool classifies the status of anxiety and depression symptoms as normal (0–7), borderline (8–10), and 11–21 (abnormal) with a binary cut off point of greater than 8 (including borderline and abnormal) considered as have anxiety and depression [17].

Association between independent variables and dependent variables was assessed and its strength presents using odd ratio and 95% confidence interval. Binary logistic regression analysis was applied. All predictor variables that have an association in bivariable analysis with a p-value < 0.25 were entered into a multivariable logistic regression model. In multivariable logistic regression analysis, those variables with a p-value ≤ 0.05 were considered statistically significant.

## Data quality control

The questionnaire was first prepared in English and then translated into Amharic version by a professional translator with a health background who was a native speaker of Amharic language and fluent in English. The backward translation from Amharic to English was done by an independent translator who was a native speaker of the source language and fluent in Amharic language. Consensus on the compatibility of forward and backward translation was assured before the actual data collection activities. Data collectors and supervisors were trained for two days by the principal investigator before the actual study commenced on the objectives of the study. As part of training, the data collection tools were pre-test in 5% of the total sample at Shashemene hospital (adjacent to the study area) before the actual data collection to check the questionnaire addressed the study variables, as well as to check the extent at which questions understood by the interviewee and to identify areas for modification and correction. The principal investigator and supervisors checked the completeness and consistency of the collected data and provided early feedback to the data collectors.

Each questioner was given a code before data entry to make data processing easier. In addition, the data entry format was prepared in Epi data software based on the pre-coded questionnaire. To reduce some errors during data entry, a check file was developed (to detect and refuse some data entry mistakes). Before conducting analysis in SPSS software, data cleaning was done to check for outliers, consistency, and to verify the skip pattern was followed. Furthermore, exploratory data analysis was performed to determine the levels of missing values and the presence of multi co-linearity.

## Ethical consideration

Hawassa University's institutional review board granted ethical clearance under reference number IRB/293/12. Written informed consent was obtained from the study participant before their participation. Study participant's confidentiality and privacy were protected by excluding their names from the questionnaire and keeping their data safe in password locked computer throughout the whole process of research work. At the end of each interview, study participants received health education on COVID-19 signs and symptoms, transmission routes, and preventive measures.

## Result

### Socio-demographic characteristics

A total of 422 participants were included in the study, which makes the response rate 100%. The median age of the respondents was 44 years with IQR of (33, 55) years old. Out of the total study participants, 230(54.5%) of them were married. From study participants, 114(27%) of them could not read and write while 164(38.9%) had above secondary educational status.

More than half (50.2%) of the participants were from urban areas. The average monthly income of study participants was 3451 (SD± 3003). About 114(27%) and 50(11.8%) were government and private employees respectively. More than one-third (33.9%) of the participants had a diagnosis of Diabetes mellitus, followed by Hypertension 88(20.9%) (**Table 1**).

## Knowledge of participants toward COVID-19

Out of 422 study participants, more than half (56.2%, 95% CI: 50.7–60.9) of them had a good knowledge of COVID-19, while the remaining 185 (43.8%) had poor knowledge. Most (93.8%) of the study participants were aware that the main clinical symptoms of the novel COVID-19 are fever, dry cough, shortness of breath, and myalgia. More than half (53.6%) of

**Table 1. Socio-demographic characteristics of chronic disease patients, at selected public hospitals of Sidama regional state, Ethiopia, 2020 (N = 422).**

| Variables | Frequency (%), N = 422 |
|---|---|
| **Sex** | |
| Male | 230 (54.5) |
| Female | 192 (45.5) |
| **Age (years)** | |
| 18–39 | 162 (38.4) |
| 40–61 | 194 (46.0) |
| >61 | 66 (15.6) |
| **Marital status** | |
| Married | 289 (68.5) |
| Single | 65 (15.4) |
| Divorced | 26 (6.2) |
| Widowed | 42 (10) |
| **Residence** | |
| Urban | 212 (50.2) |
| Rural | 210 (49.8) |
| **Monthly income (ETB)[b]** | |
| <1500 | 141 (33.4) |
| 1500–3000 | 106 (25.1) |
| >3000 | 175 (41.5) |
| **Occupation** | |
| Merchant | 69 (16.4) |
| Government employee | 114 (27) |
| Private employee | 50 (11.8) |
| Farmer | 94 (22.3) |
| Housewife | 71 (16.8) |
| Others | 24 (5.7) |
| **Type of chronic diseases** | |
| Diabetes mellitus | 143 (33.9) |
| Hypertension | 88 (20.9) |
| Heart disease | 84 (19.9) |
| Chronic lung disease | 99 (23.5) |
| Other* | 8 (1.9) |

*HIV, Gout
[b]ETB, Ethiopian Birr.

**Table 2. Knowledge about COVID-19 among chronic disease patients at selected public hospitals of Sidama regional state, Ethiopia, 2020 (N = 422)].**

| S. No | Knowledge questions | Yes (%) | No (%) | I don't know (%) |
|---|---|---|---|---|
| 1 | The main clinical symptoms of COVID-19 are fever, dry cough, shortness of breath, and myalgia | **396 (93.8)** | 10 (2.4) | 16 (3.8) |
| 2 | Unlike the common cold, stuffy nose, runny nose, and sneezing are less common in persons infected with the COVID-19 virus | **243 (57.6)** | 134 (31.8) | 45 (10.7) |
| 3 | COVID-19 symptoms appear within 2–14 days | **254 (60.2)** | 35 (8.3) | 133 (31.5) |
| 4 | Currently, there is no effective treatment or vaccine for COVID-2019, but early symptomatic and supportive treatment can help most patients to recover from the infection | **347 (82.2)** | 31 (7.3) | 44 (10.4) |
| 5 | Not all persons with COVID-19 will develop severe cases. Those who are elderly, have chronic illnesses, and with suppressed immunity are more likely to be severe cases | **367 (87.0)** | 21 (5.0) | 34 (8.1) |
| 6 | Touching or shaking hands of an infected person would result in the infection by the COVID-19 virus | **396 (93.8)** | 14 (3.3) | 12 (2.8) |
| 7 | Touching an object or surface with the virus on it, then touching your mouth, nose, or eyes with the unwashed hand would result in the infection by the COVID-19 virus | **379 (89.8)** | 33 (7.8) | 10 (2.4) |
| 8 | The COVID-19 virus spreads via respiratory droplets of infected individuals through the air during sneezing or coughing of infected patients | **381 (90.3)** | 18 (4.3) | 23 (5.5) |
| 9 | Persons with COVID-19 cannot infect the virus to others if he has no any symptom of COVID-19 | 226 (53.6) | **103 (24.4)** | 93 (22.0) |
| 10 | Wearing masks when moving out of home is important to prevent the infection with COVID-19 virus | **382 (90.5)** | 28 (6.6) | 12 (2.8) |
| 11 | Children and young adults do not need to take measures to prevent the infection by the COVID-19 virus | 243 (57.6) | **147 (34.8)** | 32 (7.6) |
| 12 | To prevent the COVID-19 infection, individuals should avoid going to crowded places such as public transportations, religious places, Hospitals and Workplaces | 377 (89.3) | 34 (8.1) | 11 (2.6) |
| 13 | Washing hands frequently with soap and water for at least 20 seconds or use an alcohol based hand sanitizer (60%) is important to prevent infection with COVD-19 | **359 (85.1)** | 42 (10.0) | 21 (5.0) |
| 14 | Traveling to an infectious area or having contact with someone traveled to an area where the infection present is a risk for developing an infection | **378 (89.6)** | 34 (8.1) | 10 (2.4) |
| 15 | Isolation and treatment of people who are infected with the COVID-19 virus are effective ways to reduce the spread of the virus | **389 (92.2)** | 20 (4.7) | 13 (3.1) |
| 16 | People who have contact with someone infected with the COVID-19 virus should be immediately isolated in a proper place | **376 (89.1)** | 25 (5.9) | 21 (5.0) |

the participants reported that a person with COVID-19 cannot infect others if he has no symptoms of COVID-19. Frequent proper handwashing with soap for 20 seconds and wearing masks when leaving the house were reported as the means of protection by 359 (85.1%) and 382 (90.5%) participants, respectively (**Table 2**).

## Participants practice toward COVID-19 prevention approach

More than half of the participants (57.6%, 95% CI: 52.8–62.1) had poor practice toward COVID-19 prevention methods. Only 179 (42.4%, 95% CI: 37.9–47.2) of the study participants had a good practice. Around 63.7% of the study participants practice physical distancing by remaining 6 feet or 2 meters from others all the time. The majority (77.3%) of respondents limit physical contact such as handshaking. More than two-thirds of participants (69.7%) touch their eyes, nose, and mouth frequently with unwashed hands (**Table 3**).

Table 3. Preventive practice toward COVID-19 among chronic disease patients at selected public hospitals of Sidama regional state, Ethiopia, 2020 (N = 422).

| S. No | Practice questions | Yes (%) | No (%) |
|---|---|---|---|
| 1 | Do you participate in meetings, religious activities, events, and other social gatherings or any crowded place in areas with ongoing community transmission? | 348 (82.5) | **74 (17.5)** |
| 2 | In recent days, have you worn a mask when leaving home? | **311 (73.7)** | 111 (26.3) |
| 3 | If yes, do you touch the front of the mask when taking it off? | 305 (72.3) | **117 (27.7)** |
| 4 | Do you reuse a mask? | 350 (82.9) | **72 (17.1)** |
| 5 | Do you wash your hands with soap and water frequently for at least 20 seconds or use sanitizer/60% alcohol | **253 (60.0)** | 169 (40.0) |
| 6 | Do you touch your eyes, nose, and mouth frequently with unwashed hands? | 294 (69.7) | **128 (30.3)** |
| 7 | Do you clean and disinfect frequently touched objects and surfaces | **260 (61.6)** | 162 (38.4) |
| 8 | Do you practice "physical distancing" by remaining 6 feet or 2 meters away from others at all times? | **269 (63.7)** | 153 (36.3) |
| 9 | Do you use other workers' phones, desks, offices, or other work tools and equipment? | 296 (70.1) | **126 (29.9)** |
| 10 | Do you limit contact (such as handshakes) | **326 (77.3)** | 96 (22.7) |
| 11 | Do you eat or drink in bars and restaurants? | 229 (54.3) | **193 (45.7)** |
| 12 | Do you cover your nose and mouth during coughing or sneezing with the elbow or a tissue, then throw the tissue in the trash | **354 (83.9)** | 68 (16.1) |
| 13 | Do you prefer to stay at home, in a room with the window open during the transmission period | **357 (84.6)** | 65 (15.4) |
| 14 | Do you stay home when you were sick due to common cold-like infection during the transmission period | **268 (63.5)** | 154 (36.5) |
| 15 | Do you listen and follow the direction of your state and local authorities? | **361 (85.5)** | 61 (14.5) |

## Prevalence of depression and anxiety among patients with chronic health problems during the COVID-19 pandemic

More than one-third of the study participants 37% (95% CI 32.7, 41.5) were experiencing anxiety. While depression affected more than a quarter of the respondents, 26.8% (95% CI 22.5, 31.5) (**Fig 1**).

## Factors associated with good knowledge about COVID-19

During the initial bivariate analysis, age, marital status, educational status, place of residence, occupation and monthly income had a significant association with good knowledge at 0.25 P-value. But, after applying a multivariate logistic regression age, marital status, place of residence, and occupation remain in the final model associated with good knowledge at 0.05 P-value.

The odds of having good knowledge in the younger age group (18–39 years) were 2.36 times higher than in the older age group (>61 years). Participants in the study who were single had a 60% lower chance of having good knowledge than those who were married. The odds of having good knowledge in urban residents were 2.02 (1.15, 3.57) times higher than the counterpart (**Table 4**).

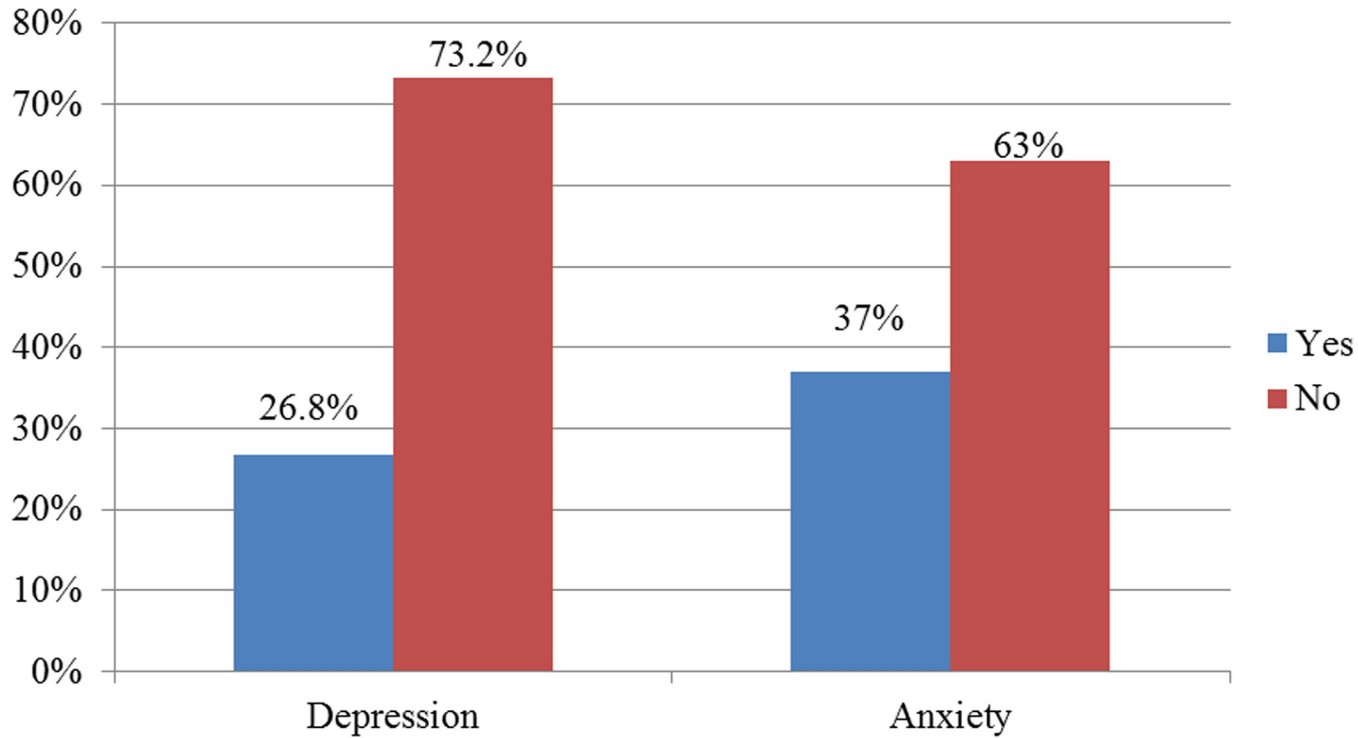

**Fig 1. Prevalence of depression and anxiety among chronic disease patients, at selected public hospitals of Sidama regional state, Ethiopia, 2020 (N = 422).**

## Factors associated with a good practice toward COVID-19 prevention approach

From the total variables entered into the multivariable regression, four variables namely marital status, educational status, place of residence, and knowledge about COVID-19 were found to be independently associated with good practice of COVID-19 prevention approaches among chronic disease patients at a P-value of <0.05.

The study participants those who were divorced and widowed were 78% and 69% (AOR = 0.22 (0.07, 0.63) & AOR = 0.31 (0.10, 0.92))) less likely to practice COVID-19 prevention approaches appropriately as compared with married, respectively. Study participants with secondary and above educational status were 2.2 times (AOR = 2.21 (1.01, 4.84) more likely to practice goodly the COVID -19 prevention methods. The odds of good practice among urban study participants were 2.3 times (AOR = 2.33 (1.30, 4.19) higher than among rural study participants. Participants who had a good understanding of COVID-19 were 4.8 (AOR = 4.87 (2.96, 8.00) times more likely to have good practices (Table 5).

## Discussion

As COVID-19 is causing a significant number of morbidity and mortality globally, putting preventive measures in action is critical to avert its impact. Improving the knowledge and practice of high-risk groups (like chronic ill patients) toward COVID-19 preventive measure are crucial for the effective prevention and control of COVID-19. Furthermore, the pandemic had an impact on the mental health of the entire population, particularly on high-risk groups. This study assessed the knowledge and practice toward COVID-19 and the mental health

**Table 4. Factors associated with good knowledge of COVID-19 among chronic disease patients, at selected public hospitals of Sidama regional state, Ethiopia, 2020 (N = 422).**

| Variables | Good knowledge | | COR (95% CI) | AOR (95% CI) |
|---|---|---|---|---|
| | Yes | No | | |
| **Age** | | | | |
| 18–39 | 101 | 61 | 2.24 (1.25, 4.02)* | **2.36 (1.12, 4.96)** |
| 40–61 | 108 | 86 | 1.70 (0.96, 2.99) | 1.19 (0.60, 2.37) |
| >61 | 28 | 38 | 1 | 1 |
| **Sex** | | | | |
| Male | 134 | 96 | 1.20 (0.82, 1.77) | |
| Female | 103 | 89 | 1 | |
| **Marital status** | | | | |
| Married | 174 | 115 | 1 | **1** |
| Single | 39 | 26 | 0.99 (0.57, 1.71)* | **0.40 (0.19, 0.81)** |
| Divorced | 17 | 9 | 1.24 (0.53, 2.89)* | **0.85 (0.32, 2.25)** |
| Widowed | 7 | 35 | 0.13 (0.05, 0.30)* | **0.23 (0.09, 0.59)** |
| **Educational status** | | | | |
| Unable to read & write | 41 | 73 | 1 | 1 |
| Read & write | 51 | 32 | 2.83 (1.58, 5.09)* | 1.63 (0.80, 3.32) |
| Elementary | 28 | 33 | 1.51 (0.80, 2.84) | 0.91 (0.42, 1.95) |
| Secondary & above | 117 | 47 | 4.43 (2.66, 7.38)* | 1.54 (0.70, 3.41) |
| **Residence** | | | | |
| Urban | 148 | 64 | 3.14 (2.10, 4.69)* | **2.02 (1.15, 3.57)** |
| Rural | 89 | 121 | 1 | **1** |
| **Occupation** | | | | |
| Housewife | 48 | 23 | 1 | **1** |
| Government employee | 88 | 26 | 7.06 (3.64, 13.69)* | **4.80 (1.73, 9.30)** |
| Private employee | 31 | 19 | 3.40 (1.59, 7.26)* | **2.94 (1.14, 7.56)** |
| Farmer | 46 | 48 | 2.00 (1.05, 3.79)* | 2.13 (1.04, 4.37) |
| Merchant | 35 | 34 | 2.14 (1.08, 4.26)* | 2.21 (0.91, 5.36) |
| Other [a] | 14 | 10 | 2.92 (1.12, 7.56)* | 2.80 (0.91, 8.54) |
| **Monthly income** | | | | |
| <1500 | 68 | 73 | 1 | 1 |
| 1500–3000 | 52 | 54 | 1.03 (0.62, 1.71) | 0.55 (0.28, 1.05) |
| >3000 | 117 | 58 | 2.16 (1.37, 3.41)* | 0.72 (0.37, 1.40) |

Notes

*P-value less than 0.25 in the bivariate analysis, 1 indicates reference category, bold numbers are P-value <0.05 in the multivariate analysis

[a] daily laborer.

impact of the pandemic (depression and anxiety) among chronic ill patients in three hospitals in the Sidama regional state.

Our finding indicates that there was a high prevalence of poor knowledge (43.8%) among chronically ill patients. This finding is higher than that of studies conducted in China and Kenya [18, 19]. This difference might be because China and Kenya surveys included urban study participants, whereas our study included nearly half (49.8%) of study respondents from rural areas. In addition, another reason might be due to the fact that 82.4% of study participants held an academic degree or associate's degree or above in China.

**Table 5. Factors associated with good practice of COVID-19 prevention methods among chronic disease patients, at selected public hospitals of Sidama regional state, Ethiopia, 2020 (N = 422).**

| Variables | Good practice | | COR (95% CI) | AOR (95% CI) |
|---|---|---|---|---|
| | Yes | No | | |
| **Sex** | | | | |
| Male | 134 | 96 | 1.23 (0.83, 1.82) | |
| Female | 103 | 89 | 1 | |
| **Age** | | | | |
| 18–39 | 101 | 61 | 1.57 (0.86, 2.82)* | 0.82 (0.36, 1.86) |
| 40–61 | 108 | 86 | 1.36 (0.76, 2.44) | 0.83 (0.39, 1.74) |
| >61 | 28 | 38 | 1 | 1 |
| **Marital status** | | | | |
| Married | 174 | 115 | 1 | 1 |
| Single | 39 | 26 | 0.59 (0.34, 1.04)* | 0.52 (0.26, 1.07) |
| Divorced | 17 | 9 | 0.37 (0.15, 0.92)* | **0.22 (0.07, 0.63)** |
| Widowed | 7 | 35 | 0.13 (0.05, 0.36)* | **0.31 (0.10, 0.92)** |
| **Educational status** | | | | |
| Unable to read & write | 37 | 77 | 1 | 1 |
| Read & write | 37 | 46 | 1.67 (0.93, 3.00)* | 1.08 (0.50, 2.32) |
| Elementary | 22 | 39 | 1.17 (0.61, 2.25)* | 1.06 (0.47, 2.40) |
| Secondary & above | 83 | 81 | 2.13 (1.29, 3.50)* | **2.21 (1.01, 4.84)** |
| **Residence** | | | | |
| Urban | 105 | 107 | 1.80 (1.22, 2.66)* | **2.33 (1.30, 4.19)** |
| Rural | 74 | 136 | 1 | 1 |
| **Occupation** | | | | |
| House wife | 28 | 43 | 1 | 1 |
| Government employee | 61 | 53 | 1.76 (0.96, 3.22)* | 0.22 (0.07, 0.64) |
| Private employee | 19 | 31 | 0.94 (0.44, 1.97) | 0.20 (0.07, 0.59) |
| Farmer | 38 | 56 | 1.04 (0.55, 1.95) | 0.64 (0.30, 1.35) |
| Merchant | 24 | 45 | 0.81 (0.41, 1.62) | 0.31 (0.11, 0.81) |
| Other [a] | 9 | 15 | 0.92 (0.35, 2.39) | 0.60 (0.20, 1.84) |
| **Knowledge about COVID-19** | | | | |
| Good | 140 | 97 | 5.40 (3.48, 8.37)* | **4.87 (2.96, 8.00)** |
| Poor | 39 | 146 | 1 | 1 |
| **Monthly income** | | | | |
| <1500 | 52 | 89 | 1 | 1 |
| 1500–3000 | 43 | 63 | 1.16 (0.69, 1.95) | 1.60 (0.80, 3.19) |
| >3000 | 84 | 91 | 1.58 (1.01, 2.48)* | 1.76 (0.85, 3.62) |

Notes

*P-value less than 0.25 in the bivariate analysis, 1 indicates reference category, bold numbers are P-value <0.05 in the multivariate analysis

[a] daily laborer.

According to this study, younger age groups were more likely to have a good knowledge than the elders. This finding is supported by the study conducted in Egypt [20] and Chicago [21], which reported that younger respondents showed good knowledge. This might be due to the physiological changes during aging. As age increases, the visual performance and hearing ability get decrease. As a result, it is difficult to read and understand various health-related issues, resulting in a lack of knowledge.

Urban residents were two times more likely to have good knowledge than rural residents. This finding is in line with the study conducted in Addis Zemen and China [15, 18]. This is because of better access to the information in urban areas, where there is better access to online Media to update themselves about COVID-19. The vast majority of rural people may lack access to the Internet and social media and rely solely on television and radio for information; which may limit their ability to learn about the disease. A study conducted in Iran showed that participants who got their information from social media, scientific articles, and journals had significant higher knowledge of the disease as compared to news media users who had significantly lower knowledge regarding the transfer routes and groups at higher risk regarding COVID-19 [22]. In addition, the majorities of rural Ethiopians are illiterate and live far from the healthcare facilities where they can get health-related information.

In this study, government employee and private employees were around 5 times and 2 times more likely to have good knowledge than housewives, respectively. This is in line with the study conducted in Jimma [23]. This is because they had strict instructions about COVID-19 infection control and preventive measures at offices compared to housewives. A chance to have a better social network at the workplace might help them in obtaining information about COVID-19.

In this study, the prevalence of good COVID-19 practice among chronically ill patients was very low (42.4%), which was in line with a finding in Ethiopia [24]. However, it was lower than the finding in Saudi Arabia (81%) [25] and Rwanda (90%) [26]. The possible justification for this disparity might be knowledge differences in the study population, attitudes toward disease, different data collection periods, and the policy of countries toward COVID-19 prevention measures at the time of data collection.

The majority (82.5%) of study participants attend social gathering events and religious activities; however, in China, and Chicago, only a small number of study participants visit any crowded place [18, 21]. This difference is due to the socio-cultural and religious differences in the study population. Measures taken by the government to prevent the transmission of COVID-19 may also be another possible difference. In response to the preventive measures being followed by the study participants, 73.3% claimed that they were using a face mask and 60% of participants said they were washing their hands with soaps and water frequently.

Study participants with secondary and above educational status were more likely to have a good practice toward COVID-19 prevention approaches than those with the educational status of unable to read and write. As with this finding, a study carried out in Iran and Adiss Zemen showed that a higher level of education was associated with a high practice of COVID-19 prevention [15, 22]. People with better educational status have a better chance of acquiring information regarding the COVID-19 prevention techniques. In addition, different studies showed that educated people have positive attitudes regarding how people should behave toward COVID-19 [19, 21, 22]. Having adequate information regarding COVID-19 prevention techniques with a positive attitude toward it leads to good practice.

The odds of good practice among divorced and widowed were 78% and 69% less likely as compared to the married one. The finding agreed with a study conducted in Pakistan [27]. This might be because married people practice COVID-19 prevention practice as they feared the spread of the disease to their partner.

The odds of good practice among urban study participants were 2.3 times higher than among rural study participants. This finding is consistent with the study done in Jimma and China [18, 23]. This might be as different studies showed that people from urban areas have a better knowledge of COVID-19 prevention, because health information that improves the knowledge and practice is becoming more accessible online, however it is not reachable to rural residents [28]. In addition, it might be due to the prevention actions set by governments being implemented well in urban areas as compared to rural.

Patients with good knowledge were more likely to practice well. This finding is in line with the study conducted in China, Adiss Zemen and Adiss Abeba [15, 18, 29]. This might be due to knowing the seriousness and infectiousness of the virus pushing them to practice the preventive measures.

The prevalence of depression (5.73%) and anxiety (32%) among patients living with a chronic medical condition in Ethiopia, before the emergence of COVID-19 [30] was found to be lower compared to the finding of the current study conducted amid the COVID-19 pandemic, which revealed a high prevalence of depression (26.8%) and anxiety (37%). The finding was lower as compared with the findings of Metu referral hospital, which reports 55.7% and 61.8% have depression and anxiety respectively [8]. This could be due to the time of data collection and the tool used to measure the variables.

## Limitations of the study

The study has certain limitations; firstly respondents might give socially acceptable answers, for example, practice-related questions were collected through participants' responses rather than through observation of what they do. In addition, the cross-sectional nature of the study did not allow us to show the cause-effect relationship. Moreover, since the study was conducted in a health facility there is a possibility of bias as the underprivileged population may not have been able to participate in the study.

## Conclusion

Insufficient knowledge about COVID-19 still exists among chronic disease patients, which could pose a barrier to good prevention practices. Age (younger age group), marital status, urban residence, and occupation of government employees were significantly associated with good knowledge. Secondary and above educational status, urban residence, and good knowledge were significantly associated with good practice related to COVID-19 prevention practice. In addition, the extent of co-occurring depression and anxiety in this study was high.

The minister of health, regional health bureau, zonal and district health offices should devise appropriate interventions to fill knowledge and practice gaps among chronic ill patients. Parallel to this, special attention should be given to those with low educational attainment who are unable to read or write, as well as the rural community. In patients with chronic health conditions, strategies for early detection and treatment of depression and anxiety should be developed. In addition, other studies should be conducted to identify the predictors of anxiety and depression.

## Supporting information

**S1 File. English version survey questionnaire.**
(DOCX)

**S2 File. Raw SPSS data.**
(SAV)

**S3 File.**
(DOCX)

## Acknowledgments

The authors would like to express their deepest acknowledgment to Hawassa University for approval of ethical clearance. The authors also would like to extend a special thanks to data

collectors, supervisors, and study participants. Finally, our special thanks go to Arsema Abebe for her guidance and support.

## Author Contributions

**Conceptualization:** Yilkal Simachew.

**Data curation:** Yilkal Simachew.

**Formal analysis:** Yilkal Simachew, Amanuel Ejeso.

**Funding acquisition:** Yilkal Simachew.

**Investigation:** Yilkal Simachew.

**Methodology:** Yilkal Simachew, Amanuel Ejeso, Sisay Dejene.

**Project administration:** Yilkal Simachew.

**Resources:** Yilkal Simachew.

**Software:** Yilkal Simachew, Amanuel Ejeso.

**Supervision:** Yilkal Simachew, Amanuel Ejeso, Sisay Dejene, Mohammed Ayalew.

**Validation:** Yilkal Simachew, Mohammed Ayalew.

**Visualization:** Yilkal Simachew.

**Writing – original draft:** Yilkal Simachew, Mohammed Ayalew.

**Writing – review & editing:** Yilkal Simachew, Amanuel Ejeso, Sisay Dejene.

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
