## [Decision Letter · Decision Letter 0]

14 Dec 2021

PONE-D-21-17635Knowledge, practice and impact of COVID-19 on mental health among patients with chronic health condition at selected hospitals of sidama regional state, EthiopiaPLOS ONE

Dear Dr. hunegnaw,

Thank you for submitting your manuscript to PLOS ONE. After careful consideration, we feel that it has merit but does not fully meet PLOS ONE’s publication criteria as it currently stands. Therefore, we invite you to submit a revised version of the manuscript that addresses the points raised during the review process.

The manuscript has been evaluated by three reviewers, and their comments are available below.

The reviewers have raised a number of concerns that need attention, and they request additional information on methodological aspects of the study and analyses. The reviewers additionally request that the manuscript be thoroughly revised for grammar and typographical errors.

Could you please revise the manuscript to carefully address the concerns raised?

We look forward to receiving your revised manuscript.

Kind regards,

Vanessa Carels

Staff Editor

PLOS ONE

Journal Requirements:

Reviewers' comments:

**Comments to the Author**

1. Is the manuscript technically sound, and do the data support the conclusions?

Reviewer #1: Yes

Reviewer #2: Yes

Reviewer #3: Partly

2. Has the statistical analysis been performed appropriately and rigorously? 

Reviewer #1: Yes

Reviewer #2: Yes

Reviewer #3: No

3. Have the authors made all data underlying the findings in their manuscript fully available?

Reviewer #1: No

Reviewer #2: No

Reviewer #3: No

4. Is the manuscript presented in an intelligible fashion and written in standard English?

Reviewer #1: No

Reviewer #2: No

Reviewer #3: No

5. Review Comments to the Author

Reviewer #1: This study assesses the knowledge and practice of chronic patients regarding COVID-19 and the frequency of anxiety and depression in these patients. COVID-19 is a current major public health problem and it presents a global challenge especially in developing countries like Ethiopia. The choice of patients with chronic diseases is good as these patients are more affected with severe form of COVID-19. The mental health impact of COVID-19 was addressed in few studies. However, future studies are needed to assess the association between COVID-19 and mental disorders, which could not be verified from this cross-sectional study. Knowledge about COVID-19 was assessed using Yes/No leading questions. Multiple choice questions were better to be used to decrease bias. Substantial language editing is needed for correction of grammatic and typing errors.

Major issues:

1. Please add more details about sampling method. How the participants were recruited from sample frame of last month? Do they pay regular visits to the hospital?

2. Justify the use of p-value of 0.25 as cut-off for significant result. It should be 0.05 as the level of significance used is 5%.

3. Please add limitations of the study.

4. Substantial language editing is needed for correction of grammatic and typing errors.

Minor issues:

1. The introduction section is lengthy, some sentences are repeated, please revise.

2. Page 11 line 96: Sentence between brackets is repeated.

3. Page 11 line 97: The authors would rather use “Assess” their knowledge instead of “know” their knowledge.

4. Page 12 lines 113&114: No studies? (Do you mean in your country? Please clarify)

5. Last paragraph: The introduction is supposed to be written before the study is conducted. The authors of the paper declared what the results would be (poor knowledge and practice) and suggested recommendations. This part should be moved to the conclusion.

6. Page 16: Please provide a reference for Bloom’s cut off point for knowledge and practice assessment.

7. Please add a reference for HADS. Clarify if borderline and abnormal scores are considered to have depression/anxiety.

8. Table 1: add a footnote with the meaning of ETB abbreviation.

9. Table 3: please mark the correct answers.

10. 1st paragraph second line: Please modify as now vaccines and lines of treatment have been developed.

11. Line 388: Is it a selected public hospital or 3 hospitals? Please correct.

12. Lines 391 to 396: please revise the explanation as more than half of the participants lived in urban areas.

13. Page 31 line 434: mention references of the “different studies” mentioned.

14. Line 440: Language is not clear, needs correction.

15. Lines 441 to 446: This part is repeated.

Reviewer #2: Knowledge, practice and impact of COVID-19 on mental well health of chronic disease patients at selected hospitals in Sidama regional states in Ethiopia were evaluated. This is an interesting paper with great potential that needs minor revisions. The paper also needs through editing because at times the language used is rather unclear.

Detailed Comments.

Abstract.

1. In the conclusion section, author reported prevalence of good knowledge and practice to be low yet in the result section author reported more than half of the participants to have good knowledge. I suggest to be consistent in reporting the results and conclusion.

Introduction:

1. This section is excessively long. The authors should shorten this section.

2. Statement on lines 71 – 72 needs to be cited.

3. First statement on paragraph 2 (lines 76-77) can be integrated in first paragraph.

4. Statement on lines 80 – 81 needs to be referenced.

5. Paragraph on lines 86 – 99 needs to be focused on people with chronic medical conditions but not the older adults since the study is on people with chronic medications.

Methods:

1. How did you determine that one had a chronic medical condition? Was it self-reported or tests were done. This needs to be elucidated.

2. Did the study have any exclusion criteria? If yes, this needs to be stated.

3. Kindly Elaborate more on the profession of the translators. What qualification do they need to ensure accurate?

4. Should all preventative measures be given the same weight in scoring?

5. Why choose dichotomous categories for knowledge and practice as opposed to multiple categories? If choosing a dichotomous approach, why use 80% as the cut-off? Please provide justification for your approach in both cases.

Results:

1. Correct the table numbers in the descriptions to align with their titles.

Discussion:

1. Vaccines and treatments: I think you should be clearer here - i.e. There are vaccines currently approved and enrolled. There are treatments that seem to help symptoms and reduce death rates, bit there is no cure.

2. The authors should describe in detail all the biases present in the study and these appear before the conclusions.

Reviewer #3: Abstract

1. Lines 29-31 need revision

2. The following terms should be written correctly: “facility-based” and “cross-sectional”

3. What was the essence of running logistic regression for initiation of preventive behaviour of COVID-19 when that is not the main outcome variable of interest in this study?

4. The study sought to assess knowledge, practice and impact of COVID-19 on mental health of chronic disease patients. However, the study failed to actually assess effect of these variables on mental health. Thus, the bivariate and multivariate logistic regression analysis should have been done on mental health and explanatory variables (socio-demographics, knowledge, practice, and impact of COVID-19).

5. The conclusion made is not consistent with the study purpose/objective. Authors should revise.

Introduction

6. Lines 76-77 is a repeat of Lines 69-70. Authors should delete. Also, provide intext citation to support line 80-82.

7. Line 113-114 is not consistent with the study title and objective stated in the abstract. Authors should revise. Authors should also clearly start the research question or objective in this section.

Methods

8. In line 139, authors should capitalize “Sidama”. Again, instead of “It has got a total of 19…” it should instead be “It has a total of 19…” Provide citation for the population stated in Lines 144-145. Also, as earlier stated, the terms in Line 148 should be “facility-based” and “cross-sectional”.

9. Lines 149-153 needs revision to improve the text. Line 166 should be in past tense.

10. Authors need to clarify what their dependent and independent variables are. Based on the topic, the dependent variable is mental health while knowledge and practice towards COVID-19 and socio-demographics are the independent variables. There is the need for the authors to harmonize their variables through the study topic and objectives to the study variables and subsequent inferential analysis. Lines 184-187, 189-192, and 195-196 should be included in the analysis section.

11. How was confidentiality, anonymity and privacy of the participants ensured in the study? How were these addressed? Change the subsection “Ethical statement” to “Ethical consideration”

Results

12. Lines 248-256 needs revision there are lots of grammatical errors. Also, when using phrases like “more than two thirds”, “more than one third”, authors should not combine both frequency and percentage. This should be rechecked and revised throughout the entire section.

13. Revise the table titles to concise form. They are too long.

Discussion

14. There are grammatical and typographical errors in Lines 389-396. For instance, countries names have not been capitalized.

15. Also, authors failed to proffer explanations/reasons for the study observations made. The discussion is thus not exhaustive.

Conclusion

16. The conclusion made is not consistent with the study purpose/objective. Authors should revise.

17. Also, recommendations should be specific and directed/targeted at agencies/organization.

6. PLOS authors have the option to publish the peer review history of their article (what does this mean?). If published, this will include your full peer review and any attached files.

Reviewer #1: No

Reviewer #2: No

Reviewer #3: **Yes: **Farrukh Ishaque Saah

---

## [Author Response · Author response to Decision Letter 0]

19 Jan 2022

Yilkal Simachew 

Lecturer and Researcher 

Hawassa University, Ethiopia 

joemakalister123@gmail.com

Dr. Vanessa Carels

Editorial Department 

Journal of PLOS ONE

em@editorialmanager.com January 19, 2022

Subject: Revision and resubmission of manuscript (PONE-D-21-17635)

Dear Dr. Vanessa 

Thank you for giving us the opportunity to submit a revised draft of the manuscript “Knowledge, practice and impact of COVID-19 on mental health among patients with chronic health condition at selected hospitals of sidama regional state, Ethiopia”, for publication in the journal of PLOS ONE. We appreciate the time and effort that you and the reviewers dedicated to providing feedback on our manuscript and grateful for the insightful comments on and valuable improvements to our paper. We have incorporated most of the suggestions made by the reviewers. The changes are marked in the revised manuscript. 

We have included the Reviewers and Editor Comments immediately after this letter and responded to them individually, indicating exactly how we addressed each concern and describing the changes we have made. Please note that reviewers’ comments are shown in bold type and our responses in plain type. Changes to the manuscript that have been made are marked in blue color in the revised version. The removal of text at certain locations is highlighted with red color in the revised manuscript.

We hope the revised manuscript will better suit the journal of PLOS ONE, but we are happy to consider further revisions and we thank you for your continued interest in our research. 

Sincerely,

Yilkal Simachew

Lecturer and Researcher at Hawassa University, in Ethiopia 

REPLY TO ACADEMIC EDITOR’S COMMENTS

Comment 1: Please ensure that your manuscript meets PLOS ONE's style requirements, including those for file naming. The PLOS ONE style templates can be found at

Response: we have now reformatted the manuscript (font size, font style, line spacing, figure caption, table caption, reference citation and file naming) according to the guidelines and style requirements of PLOSE ONE. 

Comment 2: We note that the grant information you provided in the ‘Funding Information’ and ‘Financial Disclosure’ sections do not match. When you resubmit, please ensure that you provide the correct grant numbers for the awards you received for your study in the ‘Funding Information’ section.

Response: we apologize for the inconsistency; during resubmission we have included the source of fund, the grant number of award, name of author who received award and the role of funders. 

Comment 3: In your Data Availability statement, you have not specified where the minimal data set underlying the results described in your manuscript can be found. PLOS defines a study's minimal data set as the underlying data used to reach the conclusions drawn in the manuscript and any additional data required to replicate the reported study findings in their entirety. All PLOS journals require that the minimal data set be made fully available.

Response: During the resubmission, we have included the minimal data set in supporting information file, with file name of “S2_File.save”. 

REPLY TO REVIEWERS’ COMMENTS

Comments from reviewer # 1, major issues

Comment 1: Please add more details about sampling method. How the participants were recruited from sample frame of last month? Do they pay regular visits to the hospital? 

Response: We thank the Reviewer for the useful suggestion. According to the suggested we have revised the sampling method and explained in detail how the study participants were recruited from the sample frame. The sample frame prepared for each hospital based on the number of chronic disease patients in the last one month who attend the chronic disease follow-up clinics. Based on one month data, there was around 1376, 1100 and 900 chronic disease patients who attend the chronic disease follow-up clinic at Hawassa university comprehensive specialized hospital, Yirgalem general hospital and Leku primary hospital respectively. Proportional sample to the number of patients at each Hospital was allocated, and then the systematic sampling technique (K=8) was used to select each study participant from their respective group. We have included how the sample frame for each hospital was prepared and how the study participants were recruited from the sample frame (Revised manuscript, line #178-182).

Comment 2: Justify the use of p-value of 0.25 as cut-off for significant result. It should be 0.05 as the level of significance used is 5%.

Response: We would like to apologize for this inconvenience. We believe that this has to do with the way we write. Actually, we have used P-value of 0.05 as a cut off for significant value. However, before developing the final model (multivariable logistic regression), we have identified candidate variables by running bivariate analysis. To improve the power of the study and to avoid losing some important variables, we took all predicator variables that have association in bivariate analysis with P-value of <0.25 as candidate variables for the final model (multivariable logistic regression). We have rewritten this section to make the idea clear (Revised manuscript, line #221-225).

Comment 3: Please add limitations of the study. 

Response: Thank you for pointing this out, accordingly we have incorporated your suggestion. The points included under limitation of the study are: since practice-related questions were collected by the participants’ response, not by observation of what they do, respondents might give socially acceptable answers. In addition the cross sectional nature of study did not allow to show the cause-effect relationship. Moreover, since the study was conducted in a health

facility there is a possibility of bias as the underprivileged population may not have been able to

participate in the study. Limitations of the study including the above points added before conclusion part (Revised manuscript, line #476-482).

Comment 4: Substantial language editing is needed for correction of grammatic and typing errors.

Response: We have gone through the entire manuscript carefully to correct the grammatical and typo errors with the help of language professionals. In addition, we used “grammarly” online software to edit the spelling, grammar and language usage. 

Comments from reviewer # 1, minor issues

Comment 1: The introduction section is lengthy, some sentences are repeated, please revise.

Response: Based on the comment we have revised the introduction section. We have merged repeated sentences (Revised manuscript, line #71-72), and also rewritten some paragraphs to make it concise and clear enough to the readers (Revised manuscript, line #94-98, lime#119-126). 

Comment 2: Page 11 line 96: Sentence between brackets is repeated. 

Response: we agree with the reviewer, we mentioned who are the vulnerable groups repeatedly. So, we have removed the repeated sentence in the bracket (Revised manuscript, line #102). 

Comment 3: Page 11 line 97: The authors would rather use “Assess” their knowledge instead of “know” their knowledge.

Response: we appreciate the reviewer suggestion for more accurate wording. We have substituted “know” to “assess” (Revised manuscript, line #103).

Comment 4: Page 12 lines 113&114: No studies? (Do you mean in your country? Please clarify)

Response: Thank you for pointing out this. What we tried to show the readers was, controlling the spread of COVID-19 among vulnerable groups is one of the strategies to control the pandemic. To facilitate the prevention and control of COVID-19 among high risk groups in Ethiopia, there is an urgent need to assess their knowledge and practice of COVID-19 prevention at this critical time. Despite this fact most studies in Ethiopia targeted health professionals and the general population, but not the vulnerable groups with chronic disease patients. As suggested by reviewer to clarify, we have rewritten this section including the above points (Revised manuscript, line #119-126). 

Comment 5: Last paragraph: The introduction is supposed to be written before the study is conducted. The authors of the paper declared what the results would be (poor knowledge and practice) and suggested recommendations. This part should be moved to the conclusion. 

Response: We thank the Reviewer for the critical review and the useful suggestions. In the revised version of the manuscript, we have moved this section to conclusion part, and instead we have putted significant of the study and the aim of the study at finally paragraph of introduction part (Revised manuscript, line #121-126). 

Comment 6: Page 16: Please provide a reference for Bloom’s cut off point for knowledge and practice assessment.

Response: Thank you for the comment. We have added a reference (Revised manuscript, line #209 and 213). However since reviewer #3 commented us to move this part (which state how we analyzed and categorized knowledge and practice) to data analysis, it is moved to analysis section. 

Comment 7: Please add a reference for HADS. Clarify if borderline and abnormal scores are considered to have depression/anxiety.

Response: Thank you for the comment. We added citation for HADS (Revised manuscript, line #216). In addition, we have clarified the classification of HADS; have depression and anxiety means when the score is above 8 including both borderline and abnormal score, and normal when the score is between 0-7 (Revised manuscript, line #215-216). Based on the reviewer #3 comment we have moved this to data analysis section. 

Comment 8: Table 1: add a footnote with the meaning of ETB abbreviation.

Response: As correctly suggested by the reviewers we have added a footnote with the meaning of ETB under TABLE 1 (Revised manuscript, line #291). 

Comment 9: Table 3: please mark the correct answers.

Response: we would like to thank the reviewer for the detail comment. As suggested we have now marked the correct answers by making bold for both knowledge and practice questions at Table 2 and Table 3 (Revised manuscript, line #307 and line# 333). 

Comment 10: 1st paragraph second line: Please modify as now vaccines and lines of treatment have been developed.

Response: Thank you for pointing out this important information. Since vaccines have been developed after we submitted this research work, it needs revision as reviewer suggested. We modify accordingly (Revised manuscript, line #388-390). 

Comment 11: Line 388: Is it a selected public hospital or 3 hospitals? Please correct.

Response: As suggested we corrected, instead of saying “a selected public hospital” we corrected as “three hospitals” (Revised manuscript, line #395). 

Comment 12: Lines 391 to 396: please revise the explanation as more than half of the participants lived in urban areas.

Response: Thank you for the comment. We have revise this section, including the following points: as compared to China and Kenya the prevalence of poor knowledge in our study was high, This difference might be due to the fact that China and Kenya survey included urban study participants, but our study included around half of (49.8%) of study respondents from rural area. In addition, another reason might be due to the fact that 82.4% of study participants held an academic degree and associate’s degree and above in China. However, in our study only 38.8% of the study participants had above secondary educational status. We have revised including the above points at (Revised manuscript, line #398-401). 

Comment 13: Page 31 line 434: mention references of the “different studies” mentioned.

Response: Thank you for the comment. We now have mentioned the references (Revised manuscript, line #449). 

Comment 14: Line 440: Language is not clear, needs correction.

Response: Thank you for the comment. We have revised the language and hopefully it is clearer now (Revised manuscript, line #453-455). 

Comment 15: Lines 441 to 446: This part is repeated.

Response: Thank you for your comment. When we conducted a multivariate logistic regression for knowledge and practice, an independent variable name “place of residence” show statistical significant at P-value of less than 0.05 with both knowledge and practice. As the reviewer commented the discussion for this variable seems similar. We have now rewritten this paragraph (Revised manuscript, line #460-462). 

Comments from reviewer # 2

Abstract.

Comment 1: In the conclusion section, author reported prevalence of good knowledge and practice to be low yet in the result section author reported more than half of the participants to have good knowledge. I suggest to be consistent in reporting the results and conclusion 

Response: Thank you for the comment. As suggested by reviewer, we have revised this section. The result and conclusion are now made consistent (Revised manuscript, line #53-56). 

Introduction:

Comment 1: This section is excessively long. The authors should shorten this section.

Response: we thank the reviewer for the useful suggestion. Based on the suggestion, we have revised the introduction section. We have merged repeated ideas (Revised manuscript, line #71-72), and also rewritten some paragraphs to make it concise and clear enough to the readers (Revised manuscript, line #94-98, line# 119-126). 

Comment 2: Statement on lines 71 – 72 needs to be cited.

Response: Thank you for pointing this out. We have provided citation for the line that state the causative agent of the COVID-19, where it has been identified first and the symptoms of the disease (Revised manuscript, line #74). 

Comment 3: First statement on paragraph 2 (lines 76-77) can be integrated in first paragraph.

Response: Thank you for the suggestion. As suggested, we have integrated with the first paragraph (Revised manuscript, line #71-72). 

Comment 4: Statement on lines 80 – 81 needs to be referenced.

Response: Thank you for the comment. We added citation (which is WHO COVID-19 situational dashboard) for the statement about the number of confirmed COVID-19 cases and death globally (Revised manuscript, line #83). 

Comment 5: Paragraph on lines 86 – 99 needs to be focused on people with chronic medical conditions but not the older adults since the study is on people with chronic medications 

Response: We would like to thank the reviewer for the insightful comment. As commented we have revised the paragraph by focusing only on people with chronic medical illness. In the revised paragraph we try to show the readers why this study wants to focus on patients with chronic medical illness (Revised manuscript, line #94-98). 

Methods:

comment 1: How did you determine that one had a chronic medical condition? Was it self-reported or tests were done. This needs to be elucidated.

Response: Thank you for raising an important point here. Actually our study population was those with chronic disease who attended the chronic disease follow-up clinics at 3 hopsitals of Sidama region during the study period. We included patients who already diagnosed with the chronic medical condition and visit selected hospital for follow-up. As correctly suggested by the reviewer, we have elucidated this point by rewrite the source population and study population (Revised manuscript, line #147-153). 

Comment 2: Did the study have any exclusion criteria? If yes, this needs to be stated. 

Response: Yes, it has. We have included eligibility criteria (both inclusion and exclusion criteria) that we have used during the time of recruiting the study participants (Revised manuscript, line #157-160). 

Comment 3: Kindly Elaborate more on the profession of the translators. What qualification do they need to ensure accurate?

Response: Thank you for your suggestion. As suggested by the reviewer, we have elaborated who were participated in the translation of the questioner/tool. To maintain the cross-cultural adaptation of the tool, we were used the forward and backward translation technique. First the forward translation (from English to Amharic version) was done by professional translator with health background who was native speaker of Amharic language and fluent in English. The backward translation (from Amharic to English version) was done by independent translator who was native speaker of source language and fluent in Amharic language. As correctly suggested by reviewer we have elaborated the profession of translators and the translation process by including the above points (Revised manuscript, line #227-230). 

Comment 4: Should all preventative measures be given the same weight in scoring?

Response: Thank you; you have raised an important point here. However, before we started the analysis, the research teams also have raised the same question: is it right to give the same weight of scoring for all preventive measures? We have consulted scholars and reviewed different literatures and based on our assessment different literatures give the same weight of scoring for all preventive measures (In addition to the previously published surveys of other pandemics (Iliyasu G, Ogoina D, Otu AA, Dayyab FM, Ebenso B, Otokpa D, et al. A multi-site knowledge attitude and practice survey of Ebola virus disease in Nigeria. PLoS ONE. (2015) 10:e0135955. doi: 10.1371/journal.pone.0135955) different studies on COVID-19 gives same weight in scoring of preventive measures. In addition, scholars suggested that each preventive measure have equal capacity of preventing the COVID-19 infection. Based on the output of comprehensive literature review and scholars suggestion, we have given the same weight of scoring for all preventive measures. 

Comment 5: Why choose dichotomous categories for knowledge and practice as opposed to multiple categories? If choosing a dichotomous approach, why use 80% as the cut-off? Please provide justification for your approach in both cases. 

Response: We choose to use dichotomous categories because of the tool we have used in the study and the research question of the study. In addition we have used different published literatures as a source for categorizing our study dependent variables including (Abate H, Mekonnen CK. Knowledge, attitude, and precautionary measures towards covid-19 among medical visitors at the university of gondar comprehensive specialized hospital northwest Ethiopia. Infection and Drug Resistance. 2020;13:4355), (Feleke BT, Wale MZ, Yirsaw MT. Knowledge, attitude and preventive practice towards COVID-19 and associated factors among outpatient service visitors at Debre Markos compressive specialized hospital, north-west Ethiopia, 2020. Plos one. 2021 Jul 15;16(7):e0251708). We use 80% as the cut-off, because the original Bloom’s cut off points (80-100%, 60-79% and < 59%) were adapted and modified from the KAP study conducted on COVID-19 among Gondar, Debre Markos and Adiss Zemen. 

Results:

Comment 1: Correct the table numbers in the descriptions to align with their titles.

Response: As suggested we corrected the table numbers in the description (Revised manuscript, line #317, line #350, line #372). 

Discussion:

Comment 1: Vaccines and treatments: I think you should be clearer here - i.e. There are vaccines currently approved and enrolled. There are treatments that seem to help symptoms and reduce death rates, bit there is no cure.

Response: Thank you for the comment. During the time of our manuscript submission, vaccine was not developed. But now as correctly commented by the reviewer, different countries have been developed COVID-19 vaccine to fight against the global pandemic. Therefore, we have updated this section (Revised manuscript, line #388-390). 

Comment 2: The authors should describe in detail all the biases present in the study and these appear before the conclusions.

Response: we appreciate the reviewer suggestion. Accordingly, before the conclusion we have described the biases in the study under limitation of the study subsection including the following points: since practice-related questions were collected by the participants’ response, not by observation of what they do, respondents might give socially acceptable answers. In addition the cross sectional nature of study did not allow to show the cause-effect relationship. Moreover, since the study was conducted in a health facility there is a possibility of bias as the underprivileged population may not have been able to participate in the study. (Revised manuscript, line #476-482).

Comments from reviewer # 3

Abstract

Comment 1: Lines 29-31 need revision

Response: Thank you for the comment. The sentence has been revised (Revised manuscript, line #29-30). 

Comment 2: The following terms should be written correctly: “facility-based” and “cross-sectional”

Response: We thank the reviewer for the suggestion. The term has been corrected throughout the document (Revised manuscript, line #35, line #146). 

Comment 3: What was the essence of running logistic regression for initiation of preventive behaviour of COVID-19 when that is not the main outcome variable of interest in this study?

Response: We appreciate the comment of reviewer. However, as we mentioned the general objective of the study was to assess knowledge, practice and impact of COVID-19 on mental health among chronic disease patients. The specific objective was to measure the level of COVID-19 related knowledge and to identify the factors that affect their knowledge. The second specific objective was to measure the COVID-19 prevention practice among patients with chronic medical condition and to identify the factors that affect their COVID-19 prevention practice. The other is to assess the mental health status of patients with chronic medical condition at the time of pandemic (specifically to assess anxiety and depression). We have conducted logistic regression for prevention practice of COVID-19, because one of the specific objectives of the study was to identify the factors that affect the prevention practice of COVID-19 among chronic disease patients. We agree with the reviewer, we didn’t explain the objective of the study clearly and specifically. To make it clear we have mentioned the aim of the study specifically at the end on introduction section (Revised manuscript, line #123-126). 

Comment 4: The study sought to assess knowledge, practice and impact of COVID-19 on mental health of chronic disease patients. However, the study failed to actually assess effect of these variables on mental health. Thus, the bivariate and multivariate logistic regression analysis should have been done on mental health and explanatory variables (socio-demographics, knowledge, practice, and impact of COVID-19).

Response: We would like to apologize for this inconvenience. We believe that this has to do with the way we write the study objective. As we have mentioned on our response for comment 3, we have now re-written the objective of study clearly and specifically (Revised manuscript, line #123-126). Our study objective was to assess the level of knowledge, prevention practice of COVID-19 and the factors that affect it. In addition, our study assesses the mental health status of patients with chronic medical condition during the time of COVID-19 pandemic (specifically measure anxiety and depression level). So, we used simple descriptive statistics to assess mental health, but we have used bivariate and multivariate logistic regression for knowledge and practice. 

Comment 5: The conclusion made is not consistent with the study purpose/objective. Authors should revise. 

Response: We agree with the reviewer. We now have revised and included the impact of COVID-19 pandemic on mental health as well, which is the level of anxiety and depression during this pandemic among patients with chronic medical illness (Revised manuscript, line #55-56).

Comment 6: Lines 76-77 is a repeat of Lines 69-70. Authors should delete. Also, provide intext citation to support line 80-82. 

Response: Thank you for the comment. As suggested we have deleted (Revised manuscript, line #78-79). In addition we have provided intext citation to support the line that state the number of confirmed COVID-19 cases and death globally and at Africa region (Revised manuscript, line #82). 

Comment 7: Line 113-114 is not consistent with the study title and objective stated in the abstract. Authors should revise. Authors should also clearly start the research question or objective in this section.

Response: Thank you for pointing this out. As correctly suggested by the reviewer, we have revised this section including a clear statement of research objective (Revised manuscript, line #119-126). 

Methods 

Comment 8: In line 139, authors should capitalize “Sidama”. Again, instead of “It has got a total of 19…” it should instead be “It has a total of 19…” Provide citation for the population stated in Lines 144-145. Also, as earlier stated, the terms in Line 148 should be “facility-based” and “cross-sectional”.

Response: Thank you for the comment. Based on the comment revision have made throughout the manuscript (capitalize Sidama) and fixed the typo error (Revised manuscript, line #137 and line#139). In addition we have provided citation for the estimated population of Sidama region by 2018 (Aynalem Adugna_SNNPR_January_2021http://www.ethiodemographyandhealth.org/) (Revised manuscript, line #142). The term also corrected as commented early (Revised manuscript, line #146). 

Comment 9: Lines 149-153 needs revision to improve the text. Line 166 should be in past tense.

Response: we agree with the reviewer. We have revised the text and hope that it is now clearer (Revised manuscript, line #147-154). In addition, we have rewritten line 166, the new sentence read as follow “After considering a 10% non-response rate, the final sample size was 422” (Revised manuscript, line #174).

Comment 10: Authors need to clarify what their dependent and independent variables are. Based on the topic, the dependent variable is mental health while knowledge and practice towards COVID-19 and socio-demographics are the independent variables. There is the need for the authors to harmonize their variables through the study topic and objectives to the study variables and subsequent inferential analysis. Lines 184-187, 189-192, and 195-196 should be included in the analysis section. 

Response: we apologize for the lack of clarity in our first version of the manuscript, which have given rise to the confusion. To ensure clarity of our objective and to indicate the dependent variable of the study we have revised the aim of the study at the end of introduction section (revised manuscript, line# 123-126). 

Comment 11: How was confidentiality, anonymity and privacy of the participants ensured in the study? How were these addressed? Change the subsection “Ethical statement” to “Ethical consideration”

Response: we thank the reviewer for pointing this out. Accordingly, we have included a sentence which describes how the confidentiality, anonymity and privacy of study participants were maintained. We added the following sentence to (revised manuscript, line# 251-253) in the Ethical consideration: “study participants confidentiality and privacy were maintained by excluding their name from the questionnaire and keeping their data safe in password locked computer throughout the whole process of research work”. In addition, as suggested we have changed [Ethical statement] to [Ethical consideration] (revised manuscript, line# 248).

Results

Comment 12: Lines 248-256 needs revision there are lots of grammatical errors. Also, when using phrases like “more than two thirds”, “more than one third”, authors should not combine both frequency and percentage. This should be rechecked and revised throughout the entire section.

Response: Based on the reviewer comment, we have revised and rewritten the socio-demographic part under result section (revised manuscript, line# 269-276). In addition, as suggested we have revised throughout the result section and used only percentage when we use descriptive phrases (revised manuscript, line# 296, 298, 299, 312, 315, 316).

Comment 13: Revise the table titles to concise form. They are too long 

Response: Thank you for the suggestion. We have revised the table titles and hope that it is now concise. We have revised table 2 title from [Knowledge on symptoms, transmission, prevention and practice towards COVID 19 among Chronic Disease Patients, at selected public hospitals of Sidama regional state, Ethiopia, 2020 (N=422)] to [Knowledge about COVID-19 among chronic disease patients at selected public hospitals of Sidama regional state, Ethiopia, 2020 (N=422)] (revised manuscript, line#306-307).

We have revised table 3 title from [Frequency of Response by Chronic Disease Patients for Practice Questions, at selected public hospital of Sidama regional state, Ethiopia, 2020(N=422)] to [Preventive practice toward COVID-19 among chronic disease patients at selected public hospitals of Sidama regional state, Ethiopia, 2020 (N=422)] (revised manuscript, line# 332-333).

Discussion

Comment 14: There are grammatical and typographical errors in Lines 389-396. For instance, countries names have not been capitalized.

Response: We like to thank the reviewer for detail comment to improve our research work. According to the comment we have revised and rewritten the paragraph with the help of professional language experts. The revised paragraph can be found at (revised manuscript, line# 397-401). 

Comment 15: Also, authors failed to proffer explanations/reasons for the study observations made. The discussion is thus not exhaustive.

Response: Thank you again for the insightful comment on our research work. We have revised the discussion part with a brief explanation of the study findings by relate the findings to similar studies and discussed the implications (revised manuscript, line#398-401, 409-412, 453-455, 460-462). 

Conclusion

Comment 16: The conclusion made is not consistent with the study purpose/objective. Authors should revise.

Response: Thank you for this excellent observation. We believed the way we wrote the study objective at first version of the manuscript was unintentionally misleading. Since we have rewritten the objective clearly and specifically at the revised version (revised manuscript, line#123-126), we hope the conclusion made is consistent with study objective. 

Comment 17: Also, recommendations should be specific and directed/targeted at agencies/organization.

Response: Thank you for the suggestion. Accordingly we have revised the recommendation. We hope the recommendation is specific and targeted to the responsible bodies (revised manuscript, line#491-497).

---

## [Decision Letter · Decision Letter 1]

4 Apr 2022

PONE-D-21-17635R1Knowledge, practice and impact of COVID-19 on mental health among patients with chronic health condition at selected hospitals of sidama regional state, EthiopiaPLOS ONE

Dear Dr. hunegnaw,

Thank you for submitting your revised manuscript to PLOS ONE. The three reviewers have accepted all your responses and revisions. The manuscript, however, will benefit from English editing by either a native speaker, someone proficient in English, or a professional editing service. Please submit your revised manuscript by May 19 2022 11:59PM. If you will need more time than this to complete your revisions, please reply to this message or contact the journal office at plosone@plos.org. Please include the following items when submitting your revised manuscript:A marked-up copy of your manuscript that highlights changes made to the original version. You should upload this as a separate file labeled 'Revised Manuscript with Track Changes'.An unmarked version of your revised paper without tracked changes. You should upload this as a separate file labeled 'Manuscript'.If applicable, we recommend that you deposit your laboratory protocols in protocols.io to enhance the reproducibility of your results. Protocols.io assigns your protocol its own identifier (DOI) so that it can be cited independently in the future. For instructions see: https://journals.plos.org/plosone/s/submission-guidelines#loc-laboratory-protocols. Additionally, PLOS ONE offers an option for publishing peer-reviewed Lab Protocol articles, which describe protocols hosted on protocols.io. Read more information on sharing protocols at https://plos.org/protocols?utm_medium=editorial-email&utm_source=authorletters&utm_campaign=protocols.

We look forward to receiving your revised manuscript.

Kind regards,

Chong Chen

Academic Editor

PLOS ONE

Journal Requirements:

Reviewers' comments:

Reviewer's Responses to Questions

**Comments to the Author**

1. If the authors have adequately addressed your comments raised in a previous round of review and you feel that this manuscript is now acceptable for publication, you may indicate that here to bypass the “Comments to the Author” section, enter your conflict of interest statement in the “Confidential to Editor” section, and submit your "Accept" recommendation.

Reviewer #1: All comments have been addressed

Reviewer #2: All comments have been addressed

Reviewer #3: All comments have been addressed

2. Is the manuscript technically sound, and do the data support the conclusions?

Reviewer #1: Yes

Reviewer #2: Yes

Reviewer #3: Yes

3. Has the statistical analysis been performed appropriately and rigorously? 

Reviewer #1: Yes

Reviewer #2: Yes

Reviewer #3: Yes

4. Have the authors made all data underlying the findings in their manuscript fully available?

Reviewer #1: (No Response)

Reviewer #2: No

Reviewer #3: Yes

5. Is the manuscript presented in an intelligible fashion and written in standard English?

Reviewer #1: (No Response)

Reviewer #2: No

Reviewer #3: Yes

6. Review Comments to the Author

Reviewer #1: (No Response)

Reviewer #2: Generally, the authors appropriated response to my comments and suggestions. However, some change is needed before publication:

- I believe that your manuscript could benefit by English language professional service.

Reviewer #3: Well done. All the issues raised have been sufficiently addressed. I believe the manuscript has improved greatly now.

7. PLOS authors have the option to publish the peer review history of their article (what does this mean?). If published, this will include your full peer review and any attached files.

Reviewer #1: **Yes: **Eman Anwar Sultan, Associate professor of public health, Alexandria Faculty of Medicine, Egypt

Reviewer #2: No

Reviewer #3: No

---

## [Author Response · Author response to Decision Letter 1]

14 May 2022

Yilkal Simachew 

Lecturer and Researcher 

Hawassa University, Ethiopia 

joemakalister123@gmail.com

Dr. Chong Chen

Editorial Department 

Journal of PLOS ONE

em@editorialmanager.com May 14, 2022

Subject: Revision and resubmission of manuscript (PONE-D-21-17635R1) 

Dear Dr. Chong

Thank you for giving us the opportunity to revise and resubmit the manuscript “Knowledge, practice and impact of COVID-19 on mental health among patients with chronic health conditions at selected hospitals of Sidama regional state, Ethiopia”, for publication in the journal of PLOS ONE. We wish to express our appreciation to the editor and the anonymous reviewers for your in-depth comments, suggestion, and corrections, which have greatly improved the manuscript. We have carefully considered all suggestions and concerns and tried our best to address every one of them. The changes are marked in the revised manuscript. 

We have included the Reviewers and Editor Comments immediately after this letter and responded to them individually, indicating exactly how we addressed each concern and describing the changes we have made. Please note that reviewers’ comments are shown in bold type and our responses in plain type. Changes to the manuscript that have been made are marked in blue color in the revised version. The removal of text at certain locations is highlighted with red color in the revised manuscript.

We thank you for your continued interest in our research. 

Sincerely,

Yilkal Simachew

Lecturer and Researcher at Hawassa University, in Ethiopia 

REPLY TO ACADEMIC EDITOR’S COMMENTS

Comment 1: Thank you for submitting your revised manuscript to PLOS ONE. The three reviewers have accepted all your responses and revisions. The manuscript, however, will benefit from English editing by either a native speaker, someone proficient in English, or a professional editing service.

Response: Thank you for considering our revised paper for further revision, we would like to thank all reviewers for their time and valuable comments to improve our paper. We thank you for advising us to make edition to the manuscript's English writing. A native speaker edited the manuscript’s language. Based on the report of edition, we have made changes throughout the manuscript. The changes that have made found in revised manuscript with track changes. 

Comment 2: A marked-up copy of your manuscript that highlights changes made to the original version. You should upload this as a separate file labeled 'Revised Manuscript with Track Changes'.

Response: We have uploaded a marked-up copy named “Revised Manuscript with Track Changes”.

Comment 3: An unmarked version of your revised paper without tracked changes. You should upload this as a separate file labeled 'Manuscript'.

Response: We have uploaded revised paper named “Manuscript”.

Comment 4: Response: We have not made any changes to financial disclosure.

Comment 5: Guidelines for resubmitting your figure files are available below the reviewer comments at the end of this letter.

Response: Thank you for your suggestion. Based on the suggestion, we have registered with PACE and uploaded the figure to see if it met the PLOS ONE requirements. Following the PACE report, we corrected the resolution, dimension, file format, and file size of figure. Now, the revised figure does fit with PLOS. We downloaded the figure from the PACE and uploaded as Fig 1, in TIFF format. 

Comment 6: If applicable, we recommend that you deposit your laboratory protocols in protocols.io to enhance the reproducibility of your results. Protocols.io assigns your protocol its own identifier (DOI) so that it can be cited independently in the future. For instructions see: https://journals.plos.org/plosone/s/submission-guidelines#loc-laboratory-protocols.

Response: Our study does not have any laboratory protocol to deposit.

Comment 7: Please review your reference list to ensure that it is complete and correct. If you have cited papers that have been retracted, please include the rationale for doing so in the manuscript text, or remove these references and replace them with relevant current references. Any changes to the reference list should be mentioned in the rebuttal letter that accompanies your revised manuscript. If you need to cite a retracted article, indicate the article’s retracted status in the References list and also include a citation and full reference for the retraction notice

Response: We've updated the reference list to make it complete and accurate. We updated the references to conform to the International Committee of Medical Journal Editors' reference format (ICMJE). The following is a list of updated references to ensure that it is complete and accurate, as required by the journal.

Reference number 1: According to ICMJE citation style, it is an online paper with a DOI. As a result, we have included the entire DOI next to the volume (issue) and page numbers in the revised manuscript (Revised manuscript, line #500-502). 

Reference number 2: The responsible body was not mentioned in the previous format; this has been corrected in the revised version. According to the ICMJE, because the journal article is available in two languages (English and Chinese), the journal title should be presented in its original language (Chinese). We have made the necessary changes and added the DOI and PMID (Revised manuscript, line #505-509).

Reference number 3: The source is a website, but the accessed date is missing in the previous format. We have made the necessary changes (Revised manuscript, line #511-513). 

Reference number 4: The source is a website, but the accessed date is missing in the previous format. We have made the necessary changes (Revised manuscript, line #516-519).

Reference number 5: The order of the authors' names was incorrect in the previous format. It also lacks the journal's name. The authors' names have been rearranged, and the journal title has been abbreviated in accordance with the NLM catalogue of NCBI journals (Revised manuscript, line #522-525).

Reference number 6: The accessed date is missing in the previous format. We have now mentioned the accessed date of the website (Revised manuscript, line #531-534).

Reference number 7: According to the ICMJE citation style, only the first word and proper nouns are capitalized in the title of the article. We have made the necessary changes. Furthermore, the journal title has been abbreviated in accordance with the NLM catalogue of NCBI journals. To make the citation complete, we've added DOI and PMID after the page number (Revised manuscript, line #538-542). 

Reference number 8: In the previous format, authors are listed by their first name rather than their surname. We have now revised the author's name to include the surname first, followed by the first initial. Furthermore, the previous format lacked the journal title, which we have corrected (Revised manuscript, line #545-548).

Reference number 9: In the previous format, the journal title was written in its full extent (International Journal of Environmental Research and Public Health). We have now revised it and included the journal title in its abbreviated form as per the NLM catalogue of NCBI journals (Int J Environ Res Public Health). In addition, the DOI and PMID have been updated (Revised manuscript, line #552-555).

Reference number 11: The accessed date is missing in the previous format. We have now mentioned the website's access date (Revised manuscript, line #566-569). 

Reference number 12: The author's name was not written correctly in the previous format. Furthermore, the web address was incomplete. We have now corrected the author's name and written the complete web address (Revised manuscript, line #571-572).

Reference number 14: The accessed date is missing in the previous format. We have now mentioned the website's access date (Revised manuscript, line #576-578).

Reference number 15: The arrangement of the author's name was incorrect in the previous format. We have now rearranged the author's name. Furthermore, the DOI and PMID are inserted next to the page number (Revised manuscript, line #581-584).

Page number 16: We have included the article's DOI and PMID (Revised manuscript, line #587-590).

Reference number 17: We listed all ten authors in the previous version. However, the ICMJE citation style limits the number of authors to six, followed by "et al." We made the necessary changes and added the DOI and PMID of article (Revised manuscript, line #594-597). 

Reference number 18: We limited the number of authors to six and used et al (Revised manuscript, line #601-605).

Reference number 19: The previous format did not indicate that this reference was preprint. We have now added [Preprint] next to the article title. In addition, we mentioned the citation date (Revised manuscript, line #608-612). 

Reference number 20: We limited the number of authors to six and used et al. In addition, we have included the article's DOI and PMID (Revised manuscript, line #616-620).

Reference number 21: We limited the number of authors to six and used et al. In addition, we have included the article's DOI and PMID (Revised manuscript, line #624-628).

Reference number 22: The previous format did not indicate that this reference was preprint. We have now added [Preprint] next to the article title (Revised manuscript, line #631-635).

Reference number 23: We have included the article's DOI and PMID (Revised manuscript, line #640-643).

Reference number 24: The list of authors' names has been updated. DOI and PMID have also been added (Revised manuscript, line #646-649).

Reference number 25: The journal title was misspelt in the previous format (Front Public Heal). We have made the necessary changes (Front Public Health) (Revised manuscript, line #652-656).

Reference number 26: The list of authors' names has been updated. DOI and PMID have also been added (Revised manuscript, line #659-662).

Reference number 27: We have included the article's DOI and PMID (Revised manuscript, line #665-668).

Reference number 28: The list of authors' names has been updated. The journal title was misspelt in the previous format. We have made the necessary changes. In addition, a DOI has been added (Revised manuscript, line #671-674).

Reference number 29: We have included the article's DOI and PMID (Revised manuscript, line #678-681).

Reference number 30: We have included the article's DOI and PMID (Revised manuscript, line #684-687).

Response to Reviewers' comments:

Reviewer #2

Comment 1: Generally, the authors appropriated response to my comments and suggestions. However, some change is needed before publication: I believe that your manuscript could benefit by English language professional service.

Response: Thanks for your kind comments. We appreciate your suggestion to revise the manuscript's English writing; we have addressed all concerns in this revision. The language of the manuscript was edited by a native speaker. We made changes throughout the manuscript based on the edition report. The changes found in the revised manuscript with track changes.

Reviewer #3

Comment 1: Well done. All the issues raised have been sufficiently addressed. I believe the manuscript has improved greatly now.

Response: Thank you very much.

---

## [Editor Report · Decision Letter 2]

17 May 2022

Knowledge, practice, and impact of COVID-19 on mental health among patients with chronic health conditions at selected hospitals of Sidama regional state, Ethiopia

PONE-D-21-17635R2

Dear Dr. hunegnaw,

We’re pleased to inform you that your manuscript has been judged scientifically suitable for publication and will be formally accepted for publication once it meets all outstanding technical requirements.

Kind regards,

Chong Chen

Academic Editor

PLOS ONE
---

## [Editor Report · Acceptance letter]

20 May 2022

PONE-D-21-17635R2 

Knowledge, practice, and impact of COVID-19 on mental health among patients with chronic health conditions at selected hospitals of Sidama regional state, Ethiopia 

Dear Dr. hunegnaw:

I'm pleased to inform you that your manuscript has been deemed suitable for publication in PLOS ONE. Congratulations! Your manuscript is now with our production department. 

Kind regards, 

on behalf of

Dr. Chong Chen 

Academic Editor

PLOS ONE